# An Integrative Synthetic Biology Approach to Interrogating Cellular Ubiquitin and Ufm Signaling

**DOI:** 10.3390/ijms21124231

**Published:** 2020-06-14

**Authors:** Chuanyin Li, Tianting Han, Rong Guo, Peng Chen, Chao Peng, Gali Prag, Ronggui Hu

**Affiliations:** 1State Key Laboratory of Molecular Biology, Shanghai Institute of Biochemistry and Cell Biology, Center for Excellence in Molecular Cell Science, Chinese Academy of Sciences, Shanghai 200031, China; lichuanyin2013@sibcb.ac.cn (C.L.); hantianting6@sibcb.ac.cn (T.H.); guorong2017@sibcb.ac.cn (R.G.); chenpeng@sibcb.ac.cn (P.C.); 2University of Chinese Academy of Sciences, Beijing 100049, China; pengchao@sari.ac.cn; 3National Facility for Protein Science in Shanghai, Zhangjiang Lab, Shanghai Advanced Research Institute, Chinese Academy of Science, Shanghai 201210, China; 4The Department of Biochemistry and Molecular Biology, School of Neurobiology, Biochemistry and Biophysics, George S. Wise Faculty of Life Sciences, Tel Aviv University, Tel Aviv 69978, Israel; prag@post.tau.ac.il; 5Cancer Center, Shanghai Tenth People’s Hospital, School of Medicine, Tongji University, Shanghai 20072, China

**Keywords:** post-translational modifications, UBE3A, ubiquitination, UFL1, ufmylation

## Abstract

Global identification of substrates for PTMs (post-translational modifications) represents a critical but yet dauntingly challenging task in understanding biology and disease pathology. Here we presented a synthetic biology approach, namely ‘YESS’, which coupled Y2H (yeast two hybrid) interactome screening with PTMs reactions reconstituted in bacteria for substrates identification and validation, followed by the functional validation in mammalian cells. Specifically, the sequence-independent Gateway^®^ cloning technique was adopted to afford simultaneous transfer of multiple hit ORFs (open reading frames) between the YESS sub-systems. In proof-of-evidence applications of YESS, novel substrates were identified for UBE3A and UFL1, the E3 ligases for ubiquitination and ufmylation, respectively. Therefore, the YESS approach could serve as a potentially powerful tool to study cellular signaling mediated by different PTMs.

## 1. Introduction

PTMs (post-translational modification) refer to the covalent and generally enzymatic modifications on proteins following protein biosynthesis. PTMs significantly diversify proteome and expand the chemical repertoire of genetic codons by modifying the existing functional groups or introducing new ones [1]. PTMs are specific and spatiotemporally regulated, thus conferring unique dynamics to cell signaling. Protein ubiquitination, usually through forming isopeptide or peptide bond, is one of the most commonly studied PTMs regulating almost every aspect of cellular activities [2,3,4,5,6,7,8]. Versatility of ubiquitin (Ub) signaling arises from architecturally complex poly-ubiquitin chains that individual ubiquitin (Ub) moieties may be conjugated to one or multiple residues in the substrates and Ub, while Ub itself is also subjected to other PTMs such as phosphorylation [4,9,10]. The specific interaction between E3 ligases and their respective substrates is the major factor dictating the specificity for ubiquitination modification, while the E2 interacting with the E3 that dictates the type of Ub chains [11,12]. Human genome encodes approximately 600 to 1000 putative E3 ligases that mediate ubiquitination of 70 to 80 percentages of all cellular proteins in a spatiotemporally controlled manner, leading to the hypothetically 10 to 20 substrates for each E3 ligase. Meanwhile, ubiquitin could be conjugated to a specific substrate by different E3 ubiquitin ligases, resulting into different types of Ub chain linkages that confer distinct functional consequences [10], for example sixteen E3 ligases identified so far for tumor suppressor p53 protein [13,14]. Indeed, deciphering multiplicity in the pairing of E3-substrates has constituted a long outstanding challenge to fully elucidate the pathophysiological meanings of Ub signaling.

Besides Ub, there also exist many ubiquitin like proteins (UBLs) encoded by eukaryotic genomes that can be conjugated to substrates in manners much similar to ubiquitination [15,16]. Among UBLs, the ubiquitin fold modifier 1 (Ufm1) is a recently discovered ubiquitin-like modifier that covalently attaches to substrates through a reaction termed as ufmylation, which comprises cascade of reactions mediated by specific E1-activating enzyme (UBA5), E2-conjugating enzyme (UFC1) and E3-ligating enzyme (UFL1) [17]. Ufmylation-mediated cellular signaling has received increasing attention due to its essential roles in a fast expanding spectrum of important biological processes, including erythrocyte differentiation and development, ER stress, fatty acid metabolism, biogenesis of nascent protein, maintenance of genome integrity and so forth [15,18,19,20,21]. The biological significance of ufmylation seems just to be unfolding, given the fact that ufmylation-deficient mice die embryonically in uterus and human patients with mutations in genes of ufmylation system manifested severe encephalopathy but only a few substrates have been identified so far for UFL1 [20,21,22].

Indeed, it has become one of the major themes in current biomedical research to dissect PTM-mediated cellular signaling through global identification of substrates for distinct PTM. However, this turned out to be yet practically challenging and much less efficient than desirable mainly due to the isolative, low-throughput and laborious nature of the individual steps in traditional approaches [23]. Moreover, PTMs like ubiquitination are highly dynamic, undergoing rapid turnover and constant edition through transient interactions between the substrates and conjugating or deconjugating enzymes, for example, E3 ligases or De-ubiquitinase (De-Ubs) [3]. On the contrary, although many bacterial proteins bear structural and functional resemblances to distinct components of the eukaryotic ubiquitination system, prokaryotes themselves do not bear active ubiquitination or de-ubiquitination system [24,25]. By expressing all the necessary components, ubiquitination cascades could be successfully reconstituted in bacterial system to check ubiquitination of specific E3-substrates. More recently, Prag and colleagues demonstrated that a split mDHFR (murine dihydrofolate reductase) approach, in which Ub was *N*-terminally tagged by the *N*-terminal half of DHFR and the *C*-terminal fragment of DHFR (cDHFR) fused to the *N*-terminus of the potential substrates, to detect ubiquitination events that resulted in complemented functional DHFR, which could finally support the *E. coli* cells to survive trimethoprim selection [25]. Successful reconstitution of ubiquitination system in *E. coli* highlighted a tempting potential of applying *E. coli* to study cell signaling-mediated by PTMs other than the ubiquitination modification.

Here we presented a synthetic method, namely YESS, which integrated the power of Y2H, *E. coli*, mass spectrometry as well as gene editing to afford quick and facile molecular dissection of PTMs events at high efficiency.

## 2. Results

### 2.1. The Workflow of YESS Approach

The ‘**YESS**’ approach that we presented here primarily comprised of the following steps—(1) **Y**east two hybrid (Y2H) screening to identify interacting partners using the specific ligases as baits; (2) ***E**. coli*-based system in which the PTMs reactions was faithfully reconstituted; (3) **S**ite mapping for the PTMs on the potential substrates through mass spectrometry analysis; (4) **S**ubstrate validation and characterization using the purified recombinant proteins in vitro and using the relevant mammalian cells that were genetically manipulated in vivo. (Figure 1A–D).

Firstly, this YESS system begins with Y2H screening to identify binary protein–protein interactions between ligase and their potential interacting partners (PIPs) [26], using a specific E3 ligase as the bait (Figure 1A). These PIPs in colonies that survived on the SD-4 plates and were positively blue in the X-Gal staining may just simply interact with the ligase but not necessarily be its substrates, which necessitated the subsequent PTM assays that tells whether the ligase could indeed catalyze the PTM of interests.

Secondly, if *E. coli* cells harbor no detectable activity for either catalyzing or removing the PTM of interests, the PTM could be successfully reconstituted in *E. coli* by introducing plasmids—(1) pDEST-PIPs that expressed the PIPs identified in the above Y2H screening; (2) pYESS^PTM^ that encoded the modifier, being Ub or Ufm1 and the PTM-mediating catalysts, namely, pYESS^UB^ or pYESS^UFM^ in this work (Figure 1B). A successful reconstitution of the PTMs in *E. coli* would lead to marking increases in molecular weight of the substrates, which could be readily detected by immunoblotting analysis with the recovered test proteins, using proper antibodies. Meanwhile, the tremendous advance in mass spectrometry techniques [27,28,29] has allowed not only mapping the exact sites for the modifications but fingerprinting the identity (ID) of PTM substrates (Figure 1C). This would serve as the decisive step that distinguishes the specific ligase substrates from proteins merely interacted with the ligase.

Subsequently, these newly identified substrates are subjected to individual validation and characterization assays, either through in vitro PTMs assay using the purified recombinant proteins or directly to in vivo test of their PTMs states in either *E. coli* or some relevant mammalian cells (Figure 1D). Thanks to the rapid technical advances, mammalian cells can also be genetically edited to ablate the specific ligase gene to provide a more physiological platform of intended genetic background to test the PTMs of interest. Obviously, some functional tests may also be devised and executed to further dissect the physiological roles of the ligase. Mutations at the individual PTM site of the substrate may also help to pinpoint the specific functional consequences that the PTM confers at the actual sites of the substrate, leading to full molecular dissection of PTM-mediated cellular signaling (Figure 1D).

In order to demonstrate the validity of above design and the efficiency of YESS system, next we applied it to identify and characterize the substrates for UBE3A and UFL1, which respectively representative ligases for ubiquitination and ufmylation, two major PTMs that are fundamentally important and widely interested.

### 2.2. YESS^UB^, YESS Applied to Interrogate UBE3A-Mediated Ub Signaling

UBE3A/E6AP is the founding member of the HECT (homologous to the E6AP *C*-terminal domain) family E3 ubiquitin ligases. E6AP was first named as its association with E6 protein of HPV 16 (human papillomavirus type 16) to recognize and ubiquitinate the host tumor-suppressing p53 protein, which targeted p53 for degradation [30]. In neurons, UBE3A has been shown to play a critical role in the normal development and function of the nervous system with no obvious involvement of p53. Deletion or loss-of-function mutations in human *UBE3A* gene are associated with Angelman syndrome (AS), while duplication of the chromosomal region (15q11–13) containing *UBE3A* has been discovered in 1.0–3.0% familial ASD (Autism Spectrum Disorders) worldwide [31]. UBE3A/E6AP was involved in many biological processes, such as Wnt signaling, circadian rhythms, imprinted gene networks, immunity responses, synapse plasticity and early brain development [32,33]. As UBE3A related neurodevelopmental diseases with incompletely understood mechanisms, suggesting yet undiscovered roles of UBE3A-mediated ubiquitination signal in diverse pathophysiological contexts [34,35]. In the YESS^UB^-UBE3A system, the ORF (Open reading frame) of human *UBE3A* was chosen and cloned into the pDEST32 to express the bait for Y2H screening, while prey library was constructed by cloning over ~20,000 human ORFs into pDEST22 vectors followed by large-scale transformation into pDEST32-UBE3A-containing yeast competent cells [36] (Figure 1A). Totally 27 colonies survived on the SD-4 plates and plasmids recovered from these colonies were both subjected to individual Sanger sequencing, revealing the following PIPs for UBE3A: ALDH1A2, RAD23A, PSMD4, MCM6, SERPINB2, MSTO1, IL24 and CRP in the Y2H screening (Figure 2A) and subjected to the B/P reactions of Gateway^®^ cloning system to transfer them into pDONOR vector. Subsequently, these ORFs were transferred from the pDONOR vectors to pET22b vectors using L/R reactions, generating pYESS-UBE3A-PIPs plasmids containing ORFs for expressing in *E. coli*. Meanwhile, Ha tagged ubiquitin (Ha-Ub), E1(UBA1), E2(UBH7) and E3(UBE3A) were cloned into pACYCDuet-1 vector to generate pYESS^UB^-UBE3A. In order to reconstitute ubiquitination in *E. coli* system, both pYESS^UB^-UBE3A and pYESS-UBE3A-PIPs were co-transformed into *E. coli* BL21 competent cells to allow bacterial producing all the components of ubiquitin machinery (E1, E2, E3 and Ub) and the UBE3A-interacting proteins (Figure 2B). The harvested cells (typically from 2000 mL LB culture products) were lysed in 8 M Urea lysis buffer and the cleared cell lysates were subjected to two steps of enrichments—(1) purification of all the *C*-terminal His6-tagged PIPs with Ni-NTA agarose beads in 8M Urea denaturing condition; (2) eluents from the Ni-NTA beads were immunoprecipitated with anti-HA affinity gels to enrich all Ha-Ub conjugated His6-tagged proteins. Then the immunoprecipitants were eluted, digested with trypsin and subjected to mass spectrum (MS) to identify the substrates as well as their ubiquitination sites (Figure 2C, Appendix A).

Previous work demonstrated that typical ubiquitination sites in proteins bore signature adduction of Gly-Gly to the side chain of Lys residues [37]. As showed in Figure 2C, Ub was conjugated to the side chains of 10 Lys (K) in SERPINB2 and 16 Lys (K) in ALDH1A2. The ubiquitination signals were also detected on PSMD4 and RAD23A, two known substrates identified for UBE3A previously [38,39] (Appendix A).

Furthermore, ubiquitination assays were performed with SERPINB2, ALDH1A2, PSMD4 or RAD23A in HEK293T cells. As showed in Figure 2C Gly-Gly adducts were detected on 6 Lys in SERPINB2, overlapping with those identified from *E. coli* ubiquitination system, except that on K^133^. Similarly, Gly-Gly adducts were found on 14 Lys in ALDH1A2, which were again consistent with those identified in bacterial ubiquitination system, except that on K^512^. Additionally, exactly the same ubiquitination sites were identified on RAD23A whether it was from the *E. coli* ubiquitination system or mammalian cells (Appendix A), while K^40^ and K^368^ sites were only detected on PSMD4 from mammalian cells but not those from *E. coli* (Appendix A). Our study indicated that SERPINB2 and ALDH1A2 were co-immunoprecipitated and co-localized with UBE3A in HEK293T cells (Appendix A), indicating that these two proteins could indeed form a complex with UBE3A in mammalian cells. These data suggested that the E3 ligase catalyzed ubiquitination signals seemed to be largely preserved between bacterial and mammalian systems. While for those differentially ubiquitinated sites observed here, one explanation might be that ubiquitination signals on the substrates could undergo constant edition by other E3 Ub ligases and DeUbs that are present in mammalian cells but not in *E. coli* ubiquitination system.

However, no Gly-Gly adducts were detected on any peptides derived from MCM6, MSTO1, CRP and IL24 and no ubiquitination signals were observed with them either in *E. coli* ubiquitination system or reconstituted in vitro ubiquitination assay (Figure 2D,E and Appendix A).

*UBE3A* gene was genetically ablated from multiple mammalian cells using CRISPR-CAS9 technique to generate *UBE3A^−^*^/*−*^ H1299 or *UBE3A^−/−^* SHY5Y cells [40]. The ubiquitination of SERPINB2 or ALDH1A2 was significantly reduced in *UBE3A^−BE^* H1299 cells or *UBE3A^−BE^* SHY5Y cells, compared to that in wild type cells. The ubiquitination of SERPINB2 and ALDH1A2 could be successfully reinstated by re-introduction of wild-type UBE3A but not the enzymatically dead UBE3A^C843A^ mutant (Figure 2F,G).

In mammalian cells, retinoic acid (RA) is the most important form of vitamin A metabolite and a signaling molecule that regulates and mediates multiple fundamental biological processes, including cell proliferation and differentiation and body development [41]. In RA biogenesis, ALDH1As are the rate-limiting enzymes in converting retinaldehyde to RA, which may bind to RA receptors (RARs and RXRs) and translocate into nuclear to modulate transcription of hundreds target genes whose promoter sequences typically contain one or more RA response elements (RAREs). Recently, we have discovered that excessive dosage of UBE3A gene associates with human autistic spectrum disorders (ASD) through ubiquitinating ALDH1As and down-regulating RA homeostasis [42]. As showed in Figure 2H, RARE-mediated transcription of luciferase was indeed up-regulated in *UBE3A^−BE^* SHY5Y cells compared to that in wild type cells, which could be efficiently reversed by re-introduced wild type UBE3A but not the enzymatically dead UBE3A^C843A^ mutant. To further assess the differential effect of UBE3A-mediated ubiquitination on the expression of the target genes of RA signaling, the transcriptions of endogenous *Hoxd4* and *Fgf8* gene were examined using quantitative PCR (Polymerase Chain Reaction). As shown in Figure 2I, compared with that in wildtype SHY5Y cells, the transcription of *HoxD4* or *Fgf8* was approximately 40% higher or ~50% lower, respectively, in *UBE3A^−BE^* SHY5Y cells, both of which could be almost completely reversed by re-introduction of UBE3A but not UBE3A^C843A^ (Figure 2I). These data suggested that the E3 ligase activity of UBE3A was essential for its regulatory roles in RA signaling.

Urokinase-type plasminogen activator (uPA) in conjunction with its specific cell surface receptor, uPAR, plays an important role in regulating pericellular proteolysis, while SERPINB2 or PA inhibitor-2 (PAI-2), a serine protease inhibitor of the SERPINB superfamily, is a coagulation factor that inactivates tPA and urokinase [43]. Similarly, UBE3A-mediated ubiquitination of SERPINB2 was found to negatively regulate the inhibitory effect of SERPINB2 on uPA/uPAR signaling pathway, which possibly through disrupting the interaction between SERPINB2 and uPA (Figure 2J).

### 2.3. YESS^UFM1^, YESS Applied to Identify and Characterize Substrates for UFL1-Mediated Ufmylation

Similar to ubiquitination system described above, an Y2H screening was performed using human UFL1 as the bait. As a result, sixteen UFL1 interacting proteins were identified, including previously reported substrate DDRGK1, also named as UFBP1 (UFM1-binding protein 1) [15] (Figure 3A). The pYESS^UFM1^ vector was constructed to provide the expression of human UFC1, UBA5, UFL1 and UFM1 in *E. coli* using pACYCDuet-1 as the backbone. The ORFs of these 16 genes were cloned into pET22b vector through B/P and L/R reactions in Gateway^®^ cloning system to generate pYESS-UFL1-PIPs. *E. coli* BL21 competent cells were then co-transformed with both pYESS^UFM1^ and pYESS-UFL1-PIPs to provide the expression of all components that essential for ufmylation machinery (Figure 3B), as well as UFL1-interacting proteins. The cells were lysed in 8 M Urea lysis buffer and followed by other steps in the ‘YESS’ workflow. As showed in Figure 3C, ufmylation was detected on three substrates including the previously reported DDRGK1. Specifically, UFM1 was found to be conjugated to the side chains of 4 Lys (K) on DDRGK1, 1 Lys (K) on MT1M and 1 Lys (K) on TSC22D3. Mass spectrometry graphs for UFL1-mediated ufmylation sites on MT1M and TSC22D3 were shown in Appendix A.

GST pull-down assay indicated that UFL1 directly interacted with DDRGK1, MT1M and TSC22D3 (Appendix A) and immunofluorescence microscopy analysis revealed that UFL1 co-localized with DDRGK1, MT1M and TSC22D3 in HeLa cells (Appendix A). The ufmylation of DDRGK1, MT1M and TSC22D3 could be confirmed by individual assays in *E. coli* ufmylation system, in vitro ufmylation assay and HEK293T cells that were wild type or *UFL1^−FL^* cells expressing UFL1 or not (Figure 3D–F). Therefore, MT1M and TSC22D3 were emerging as new authentic substrates for UFL1. MT1M, member 1 of metallothionein (MT) family, was shown to block the transactivation of NF-κB pathway in vivo [44]. NF-κB luciferase activity assays were performed with HEK293T cells ectopically expressing UFL1 and MT1M or its K31R mutant that was defective for UFL1-mediated ufmylation, with or without TNF-α stimulation. As shown in Appendix A, both wild type MT1M and ufmylation-defective K31R (MT1M_K31R_) inhibited TNF-α upregulated NF-κB luciferase activity, while UFL1 only disrupted the inhibitory effect of wild type MT1M on cellular NF-κB luciferase activity but had a little effect on that of MT1M_K31R_, suggesting UFL1-mediated ufmylation on K^31^ of MT1M could serve as a negative regulatory factor in controlling the inhibitory effect of MT1M on NF-κB pathway. The down regulation of MT1M and its aberrant ufmylation signaling is increasingly acknowledged as a new biomarker and potential drug target in multiple types of tumors [22,45,46]. Further studies are warranted to understand whether and how ufmylation of MT1M and the other substrates mediated signaling that contribute to the observed phenotypes in ufmylation-deficient mice and human maladies associated with aberrant Ufm1 signaling.

## 3. Discussion

Global identification of substrates for a specific PTM-catalyzing enzyme represents a yet daunting challenge task in understanding its biology and disease pathology. So far, many methods were developed to fill the unmet gaps and each has its own strength and restraints. As early as 2009, using a label-free quantitative proteomics strategy, Burande et al. successfully identified filamin A and filamin B as the first ASB2 targets to proteasome degradation [47]. These methods included recent innovative approaches including ligase trapping [48] or orthogonal UB transfer (OUT) [49] that started with engineering mutant Ub and E2–E3 pairs that were engineered to conjugate only mutant Ub but not wild-type Ub to identify potential substrates for E3 ligases of interest.

Here, in the above proof-of-concept applications of YESS approach, we have demonstrated the validity of the working principle behind YESS and its potential applications in investigating different types of PTMs. As showed above, Y2H, reconstituted *E. coli* and site mapping steps were high-throughput by their technical nature (Figure 1A–D). The Gateway^®^ cloning method was adopted to allow parallel cloning of ORFs and the transfer between expression vectors for different systems, with pDONOR as intermediate vector, pDEST22 and pYESS-PIPs for expression in yeast and *E. coli*, respectively. Obviously, this sequence-independent cloning strategy could significantly save time by transferring the ORFs without the need of knowing any information about the PIPs.

In this YESS approach, by introducing the functional components of the PTM reaction into *E. coli* that originally lacked the PTM activity, it is possible to test the enzyme-substrates pairs either individually or collectively, free of the potential interference from the other PTM-capable enzymes that exhibits in the original system, for example, other E3 ligases in the mammalian cells. With efficiency comparable to the in vitro ubiquitination/ufmylation system, this *E. coli*-based system bypassed the often painful and laborious purification of each PIP hit identified in Y2H screening and the components needed to assemble the PTM reactions in vitro. After two steps of affinity purification to enrich the ubiquitinated/ufmylated substrates, the increasingly powerful mass spectrum techniques now allow identification of multiple ubiquitination/ufmylation sites in a set of substrates. This offers one alternative but more direct and efficient ubiquitination/ufmylation assay, which easily distinguishes the PTM substrates from the mere interacting partners for a specific ligase.

Certainly, this YESS was not going without limitations. Firstly, YESS seems limited to a discrete number of E3 ligases, especially the HECT domain (homologous to the E6AP *C*-terminal domain) and some RING type domain E3 ligase but YESS approach in its current form did not accommodate those E3 Ub ligases, for example, SCF family, that only function in complex with multiple proteins [50]; the RING E3s with unknown or poorly characterized E2s and the targets (and the E3s) that do not express correctly in bacteria also not suitable for our YESS approach. Secondly, the YESS approach is that a single E2 is used at the time, while E3 ligases generally can cooperate with several E2 enzymes, which can modify the type of ubiquitin chains built on the targeted substrate; most of the RING E3s are versatile and use different E2s for different purposes and some E3s even need the presence of two E2 enzymes for efficient poly-ubiquitin chain synthesis [51]. Thirdly, due to the use of *E. coli* system, the production of conformationally-correct proteins is more an exception than the rule; ubiquitination can be occurred due to the exposure of protein surfaces that are buried in native proteins and may change an E3 partner to an E3 substrate or at least create non-physiological sites of ubiquitination. Lastly, some proteins need to be modified, such as phosphorylated or dephosphorylated, before being ubiquitinated under physiological conditions [52], while this is obviously not possible in bacteria. For future studies, validating the targets and confirming the nature of the PTM and its localization in mammalian cells will be mandatory for validating the YESS approach.

Using our YESS approach, here only one new substrate was identified for well-studied UBE3A and two new substrates for UFL1, while the sites identified by YESS might not cover all potential residues modified by ubiquitination/ufmylation. Therefore, there is plenty room left for further improvement in terms of the design and utilities of the YESS approach.

In summary, here we presented YESS as an integrative synthetic biology tool to molecularly dissect PTMs-mediated cell signaling at high efficiency and demonstrated its potential applications in study ubiquitination and ufmylation mediated by distinct ligases. Further use of YESS will not only expand our current understanding of the PTMs and related signaling events in both health and disease conditions but also facilitate therapeutic intervention into human maladies associated with dysregulated PTMs.


## 4. Materials and Methods

### 4.1. Plasmids Construction

The detailed information of plasmids used in this study is listed in Appendix A. Briefly, restriction enzyme digestion and ligation reactions (NEB) were performed using routine cloning methods. Plasmids used in Y2H screening were obtained by Gateway^®^ cloning system (Thermo Fisher, Waltham, MA, USA). Point mutations were introduced using the QuikChange site-directed mutagenesis kit (Stratagene, La Jolla, CA, USA).

### 4.2. Yeast Two-Hybrid (Y2H) Screening

Y2H screening was performed as previously described [36], using indicated E3 ligases as the baits. Firstly, pDEST32-UBE3A/UFL1 were transformed into yeast strain Mav203 (Thermo Fisher), monoclonal cells were picked and prepared for yeast competent cells, then over 20,000 human ORFs library (pDEST22 backbone) were transformed into the competent cells. The number of competent cells should be ten times more than the ORFs library and fifty 15 cm yeast culture plates were used in this study for each Y2H screening. The positive colonies survived on SD-4 (deficient in Ura, His, Leu and Trp) plates and showed a blue color in the presence of X-Gal (Sigma, St. Louis, MO, USA) were picked and cultured, then the plasmids were extracted and genes they contained were identified by first generation sequencing.

### 4.3. Co-Transformation and Two-Step Enrichment of the PTM Substrates in E. coli

Ha-Ub/UFM1, UBCH7/UFC1 and UBA1/UBA5 were assembled in the first multiple cloning sites (MCS) under T7 promoter/lac operator and ribosome-binding sites (RBS) of the dual-expression backbone pACYCDuet-1 vector (Novagen, Madison, WI, USA) using traditional cloning methods and each gene was separated with Shine-Dalgarno (SD) sequences to form a polycistronic cassette. UBE3A/UFL1 was cloned into the second MCS, generating the plasmids pYESS^UB^ or pYESS^UFM1^. The pYESS^UB^ or pYESS^UFM1^ plasmids were transformed into *E. coli* BL21 cells and competent cells were prepared. Then the pYESS-PIPs (derived from the pET22b backbone vectors) were transformed into the competent cells prepared above, cultured on ten 15 cm plates contained both chloramphenicol and ampicillin antibiotics at 37 °C for 24 h. The bacterial cells were scraped from the plates, cultured in LB (lysogeny broth) medium and subjected to induction with 0.25 mM IPTG (isopropyl β-d-1-thiogalactopyranoside, Sigma, St. Louis, MO, USA) at 16 °C for 16 h. Next day, the cells were harvested and re-suspended in 8 M Urea lysis buffer (50 mM Tris-Cl, 50 mM Na_2_HPO_4_, 0.5% NP-40, 300 mM NaCl, 8 M Urea, 20 mM imidazole, pH 8.0) for high-pressure cell disruption. The debris were pelleted by centrifugation and discarded, while the supernatants were subjected to incubation with Ni-NTA agarose beads for 4 h, washed 3 times with 8 M Urea lysis buffer and eluted with RIPA buffer (50 mM Tris-Cl, 150 mM NaCl, 5 mM EDTA (Ethylene Diamine Tetraacetic Acid), 1% NP40, 0.1% SDS, pH 7.4) supplemented with 1 M imidazole before 9 volumes of RIPA buffer were added. Then the second immunoprecipitation was performed with anti-HA affinity gel and washed three times with RIPA buffer before eluting with 8 M Urea buffer (100 mM Tris-Cl, 8 M Urea, pH 8.0). The eluents were preserved at −80 °C or directly subjected to mass spectrum analysis.

### 4.4. Ubiquitination/Ufmylation Assay in Reconstituted E. coli System

pYESS-Ha-Ub-UBCH7-UBA1/pYESS-Ha-UFM1-UFC1-UBA5, pYESS-Ha-Ub-UBCH7-UBA1-UBE3A/UBE3A (C843A) or pYESS-Ha-UFM1-UFC1-UBA5-UFL1 and pYESS-SERPINB2/ALDH1A2/MCM6/RAD23A/PSMD4/IL-4/MSTO1/CRP/MT1M/DDRGK1/TSC22D3 were co-transformed into *E. coli* BL21 competent cells, monoclonal cells were picked, cultured in LB medium and induced with 0.25 mM IPTG at 16 °C for 16 h, then subjected to two-step immunoprecipitation as described above and detected by immunoblotting assay.

### 4.5. Expression and Purification of Recombinant Proteins

GST- or His6 (His)-tagged proteins were expressed in the *E. coli* BL21 cells. After inducing with IPTG, cells were pelleted, lysed in PBS (phosphate buffer saline) buffer and incubated with glutathione or Ni-NTA beads (GE, Little Chalfont, Buckinghamshire, UK) to enrich the respective proteins, followed by elution with 25 mM L-glutathione reduced or 1M imidazole dissolved in PBS buffer (pH 8.0), then dialyzed in PBS buffer supplemented with 20% glycerol before aliquoted and preserved at −80 °C

### 4.6. In Vitro Ubiquitination/Ufmylation Assay

In vitro ubiquitination/ufmylation was performed as previously described [17,36]. Briefly, recombinant 200 ng His6-Ub, 200 ng His6-UBA1/UBA5 (E1), 250 ng His6-UBCH7/UFC1 (E2), 500 ng GST-UBE3A/UFL1 (E3) and 500 ng SERPINB2/ALDH1A2/MCM6/RAD23A/PSMD4/DDRGK1/MT1M/TSC22D3/-Flag-His6 were added into ubiquitination/ufmylation buffer (25 mM Tris-Cl, 100 mM NaCl, 1 mM DTT, 5 mM MgCl_2_, 2 mM ATP pH 7.6) with the final reaction volume of 50 µL and incubated at 37 °C for 1.5 h. The ubiquitination/ufmylation levels of proteins were detected by immunoblotting assay using anti-Flag antibody.

### 4.7. In Vivo Ubiquitination/Ufmylation Assay

Different cells were transfected with indicated plasmids or not before harvesting 48 h later, lysed in buffer A (50 mM Tris-Cl, 150 mM NaCl, 1% NP-40, 1% sodium deoxycholate, 1% SDS, PH 7.6), then boiled at 95 °C for 10 min, diluted 10 times with buffer B (50.0 mM Tris-Cl, 150.0 mM NaCl, pH 7.6) and sonicated, followed by immunoprecipitation with specific antibodies at 4 °C overnight, washed thrice with RIPA buffer (50 mM Tris-Cl, 150 mM NaCl, 1.0% NP-40, 0.5% sodium deoxycholate, 0.1% SDS, pH 7.6). The immunoprecipitants were boiled in 2× SDS-PAGE sampling buffer and subjected to immunoblotting analysis using anti-Ub or other relevant antibodies.

### 4.8. Two-Step Enrichments of the Proteins from Mammalian Cells

HEK293T cells were co-transfected with pCDNA3.0-Myc-UBE3A, pRK5-His-Ub and pCDNA3.0-SERPINB2/ALDH1A2/RAD23A/PSMD4-Flag and harvested 48 h later. Then the cells were re-suspended in 8 M Urea lysis buffer and sonicated for cell lysis. The cell debris were pelleted by centrifugation and discarded, while the supernatant was first enriched with Ni-NTA agarose beads for 4 h, washed three times with Urea lysis buffer, eluted with RIPA buffer supplemented with 1 M imidazole and 9 volume RIPA buffer were added. Then the 2nd enrichment was achieved through immunoprecipitation with anti-Flag affinity gels (Sigma, St. Louis, MO, USA), followed by washing three times with RIPA buffer and eluted with 8 M Urea buffer.

### 4.9. Mass Spectrometry Analysis to Map the Ubiquitination/Ufmylation Sites on Substrates

Recovered from the reconstituted *E. coli* ubiquitination/ufmylation system or in vivo ubiquitination assay with mammalian cells, protein samples were subjected to conventional mass spectrometry analysis as described before [53]. Briefly, the eluted proteins in 8 M Urea buffer were incubated with TCEP (Tris (2-carboxyethyl) phosphine, Sigma) for reducing potential disulfide bonds, followed by NEM alkylation and trypsin digestion. Peptides were separated on EASY-nLC system (Thermo Fisher) and analyzed by the Q Exactive mass spectrometer (Thermo Fisher), followed by identifying the modification-containing peptides using Thermo Proteome Discoverer 2.1 (Thermo Fisher) and search against Uniprot Human database.

### 4.10. Cell Culture and Transfection

HeLa, HEK293T, H1299 and SHY5Y cell lines were all cultured in Dulbecco’s modified Eagle’s medium (DMEM) supplemented with 10% FBS and 50 µg/mL penicillin/streptomycin (Life Technologies). Plasmids transfections were performed using Lipofectamine 2000 (Invitrogen) as instructed by manufacturer.

### 4.11. Generation of UBE3A/UFL1-Ablated Cell Lines

*UBE3A*^−/−^ H1299, *UBE3A*^−/−^ SHY5Y and *UFL1*^−/−^ HEK293T cells were generated by genome editing using the CRISPR/Cas9 technique as previously described [40]. The sgRNAs targeting UBE3A/UFL1 were designed using online tool (http://crispr.mit.edu/) (targeting sequences are in lowercase):
UBE3A forward primers: 5′-CACCGagcacaaaactcattcgtgc-3′,UBE3A reverse primers: 5′-AAACgcacgaatgagttttgtgctC-3′.UFL1 forward primers: 5′-CACCG ccagcgggcgcagttcgccg-3′,UFL1 reverse primers: 5′-AAAC cggcgaactgcgcccgctggC-3′.

The oligos were annealed and cloned into pLentiCRISPRv2 vector (Addgene) [54]. The sgRNA-expressing plasmids were transfected into H1299, SHY5Y or HEK293T cells with lipofectamine 2000 (Invitrogen, Carlsbad, CA, USA) for 24 h, selected for survival in medium containing 5 µg/mL puromycin (Sigma) and followed by limited dilution for the formation of single colonies. Single cell colonies were picked, amplified and subjected to immunoblotting analysis using individual antibodies. Genomic DNAs were extracted and specific target sequences were then amplified followed by Sanger sequencing to confirm the aimed editions in the genomes of each cell line.

### 4.12. Co-Immunoprecipitation and Immunoblotting Assay

For co-immunoprecipitation, cells expressing the proteins of interest were lysed in Co-IP buffer (50 mM Tris-Cl pH 7.5, 150 mM NaCl, 5.0 mM EDTA, 1.0% NP-40 pH 7.6) supplemented with protease inhibitor cocktail (Roche, Basel, Switzerland). The cleared supernatant lysates were incubated with specific antibodies and protein G agarose beads or incubated with Anti-Flag affinity gels. The immunoprecipitants were denatured at 100 °C for 10 min in 2× SDS-PAGE sampling buffer. The inputs, immunoprecipitants and other cell lysates were then subjected to SDS-PAGE and transferred to PVDF (Polyvinylidene Fluoride) membranes (Bio-Rad, Hercules, CA, USA). The membranes were blocked with 5% non-fat milk, incubated with specific primary antibodies and HRP (Horseradish peroxidase)-conjugated secondary antibodies, then finally visualized with ECL Western Blotting Reagent (Tanon, Shanghai, China) using Tanon 5200 Imaging System (Tanon). All antibodies used in this study were listed in Appendix A.

### 4.13. Immunofluorescence Microscopy

HeLa cells were co-transfected with pEGFP-UBE3A/UFL1 and pCDNA3.1-SERPINB2/MT1M/DDRGK1/TSC22D3-RFP, then the cells were fixed and stained with DAPI (4’, 6-diamidino-2-phenylindole, Sigma) 48 h later. Images were recorded with microscope BX51 (Olympus, Tokyo, Japan).

### 4.14. GST Pull-Down Assay

Purified recombinant GST-UFL1 (20µg), MT1M/DDRGK1/TSC22D3-Flag-His6 (20 µg) and Glutathione Sepharose 4B beads (GE) were incubated at 4 °C overnight in 500 µL of pulldown buffer (20 mM Tris-Cl, 100 mM NaCl, 5 mM MgCl_2_, 1 mM EDTA, 1 mM DTT, 0.5% (*v/v*) NP-40 and 10 µg/mL BSA, pH 7.5). The beads were pelleted and washed three times with the pull-down buffer (10 min incubation at 4 °C for each washing). Then the recovered beads were boiled in 2× SDS-PAGE sampling buffer and subjected to immunoblotting analysis using indicated antibodies.

### 4.15. Luciferase Assay

The plasmid pGL4.22-RARE luciferase was constructed by cloning 3× RARE (retinoic acid responsive elements) into the pGL4.22 vector (Promega, Madison, WI, USA) and the plasmid pGL3-NF-κB luciferase was constructed by cloning NF-κB-binding DNA sequence into the pGL3 vector (Promega). PRL-TK, pGL4.22-RARE/pGL3-NF-κB and other vectors were co-transfected into SHY5Y or HEK293T cells. After 48 h transfection, the cells were harvested and lysed with 5× passive buffer and then subjected to Dual-Luciferase Reporter assay according to manufacturer’s instruction (Promega).

### 4.16. Quantitative Real-Time PCR (qPCR)

Total RNAs were extracted from the indicated cells with RNeasy Plus Kit (Qiagen, Duesseldorf, Germany). Complementary DNAs (cDNAs) were synthesized using ReverTra^®^ Ace qPCR RT Master Mix (Toyobo, Tokyo, Japan). Quantitative Real-time PCR (qPCR) was performed using SYBR Green (Toyobo) on a 7500 fast real-time PCR machine (ABI) and the relative abundance of each transcript was normalized to that of *GAPDH* gene using the 2^ΔΔCt^ method. All qPCR data were presented as mean ± SD, *n* = 3. Sequences for the primers used in this study were listed in Appendix A.

### 4.17. uPA Activity Assay

The uPA activity of cell lysate from wild-type or *UBE3A*^−/−^ H1299 cells expressing UBE3A or UBE3A^C843A^ were detected using uPA activity assay kit (ECM600, Millipore, Billerica, MA, USA) according to manufacturer’s instruction. Briefly, uPA cleaves and activates the chromogenic substrate to generate a colored product, detectable by its absorbance with optical density at 405 nm (OD405).

### 4.18. Statistics

Data were analyzed by two tailed unpaired t-test or one-way ANOVA with Bonferroni post-hoc test using GraphPad Prism 7. * *p* < 0.05 was considered to be significant, ** *p* < 0.01 was considered to be very significant.

## Figures and Tables

**Figure 1 ijms-21-04231-f001:**
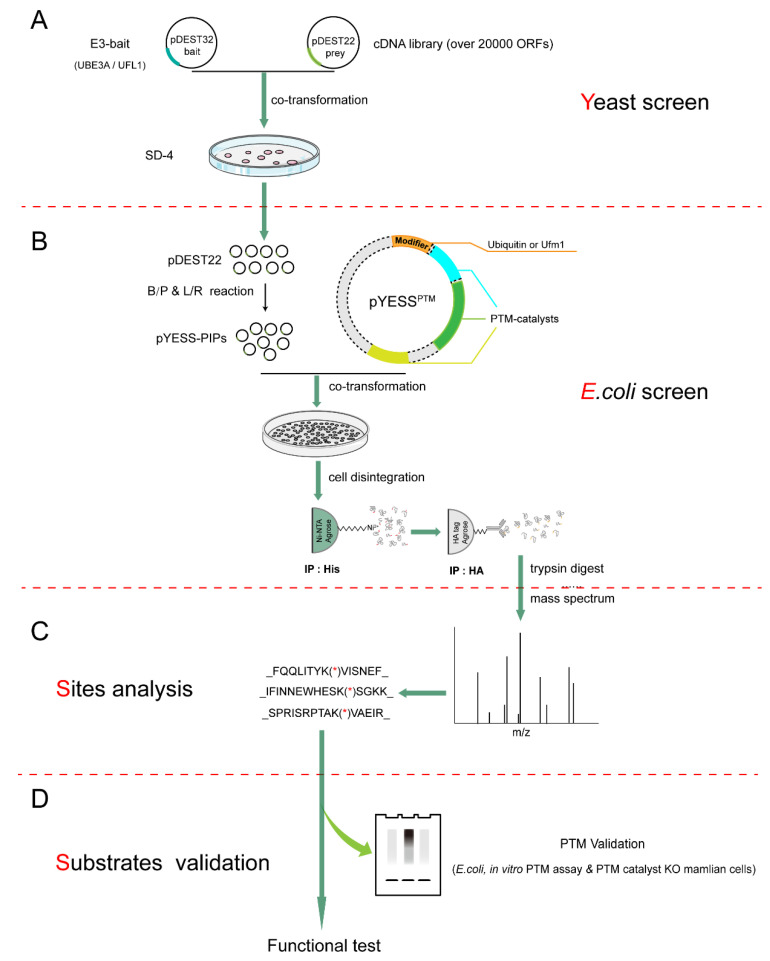
The workflow of YESS approach. Schematic illustration of the four major steps for YESS: (**A**) **Y**east two hybrid (Y2H) screening to identify interacting proteins of an E3 ligase (UBE3A/UFL1). (**B**) ***E**. coli*-based system to reconstitute ubiquitination/ufmylation followed by affinity enrichment of the modified substrates. (**C**) **S**ite mapping for ubiquitination/ufmylation by mass spectrum analysis. *, ubiquitination/ufmylation sites. (**D**) **S**ubstrates validation individually tested in ***E**. coli*, in vitro and in vivo ubiquitination/ufmylation system, which could be followed by further functional tests in relevant systems. **Y**, Yeast two-hybrid (Y2H); **E**, *E. coli* screen; **S**, Sites analysis; **S**, Substrates validation.

**Figure 2 ijms-21-04231-f002:**
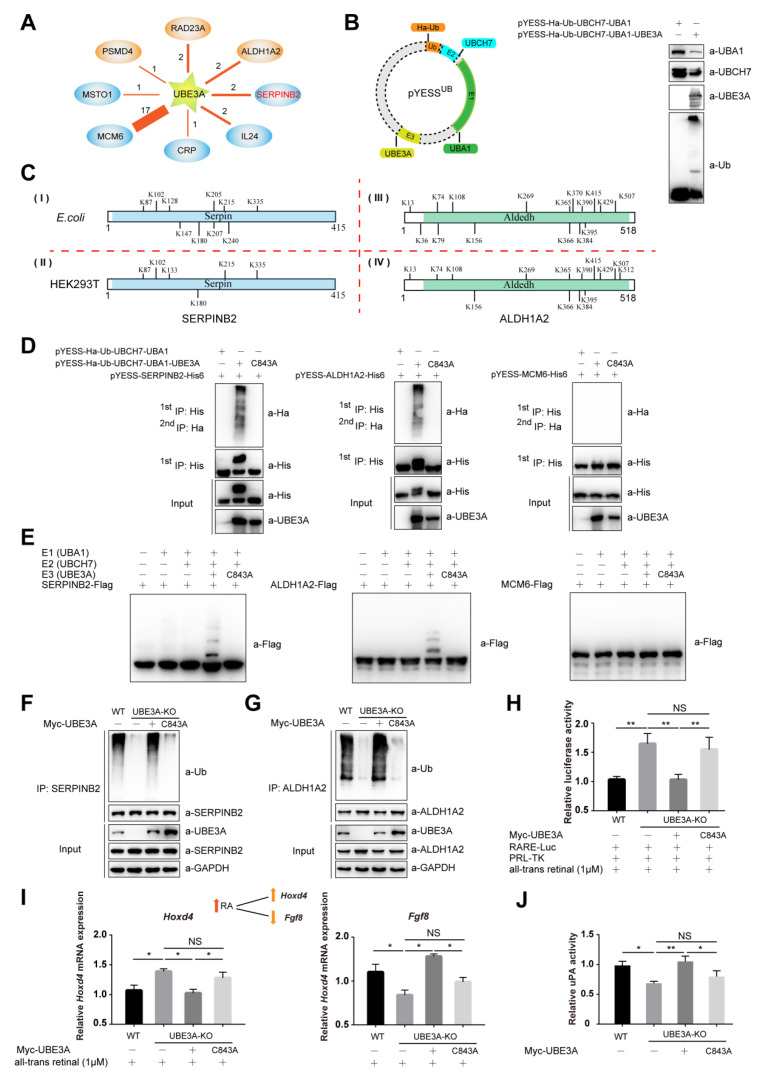
YESS^UB^, YESS applied to interrogate UBE3A-mediated Ub signaling. (**A**) UBE3A-interacting proteins screened by Y2H with human UBE3A as the bait. (**B**) Reconstituting the ***E**. coli* ubiquitination system by transforming YESS^UB^ vector that expressing Ub(Ha-Ub), E1(UBA1), E2(UBCH7) and E3(UBE3A) into BL21 cells and the expression of these proteins was detected by immunoblotting assay. (**C**) A schematic illustration of the ubiquitination sites of SERPINB2 (I and II) or ALDH1A2 (III and IV) identified in ***E**. coli* (upper panel) or mammalian HEK293T cells (lower panel), as revealed by mass spectrum analysis. (**D** and **E**) Substrates validation. UBE3A-interacting SERPINB2, ALDH1A2 and MCM6 were individually subjected to ubiquitination assay in ***E**. coli* (**D**) or in vitro (**E**) and MCM6 was negative in both tests. +, transfected with indicated plasmid; −, without transfected with indicated plasmid. (**F**) Detection the ubiquitination of SERPINB2 in wild type or *UBE3A*^−/−^ H1299 cells expressing UBE3A or the enzymatically dead C843A mutant. (**G**) Detection the ubiquitination of ALDH1A2 in wild type or *UBE3A*^−/−^ SHY5Y cells expressing UBE3A or the enzymatically dead C843A mutant. -, *UBE3A* gene knoctout allele. (**H** and **I**) Functional analysis of UBE3A mediated ubiquitination of ALDH1A2, which regulates retinoic acid homeostasis as well as its downstream signaling pathway. RARE-luciferase reporter assay (**H**) or the relative mRNA abundances of *Hoxd4* and *Fgf8*, the two targeted genes in retinoic acid signaling (**I**). Wild type or *UBE3A*^−/−^ SHY5Y cells that expressing UBE3A or the enzymatically dead C843A mutant were treated with 1.0 µM all-trans retinoic acid for 8 h before harvesting. Data are presented as mean ± SD, one-way ANOVA, with Bonferroni post-hoc test, three independent experiments. * *p* < 0.05, significant difference; ** *p* < 0.01, very significant difference; NS, no significant difference. (**J**) UBE3A-mediated ubiquitination of SERPINB2 disrupted the inhibition of SERPINB2 on uPA activity. Cell lysates from wild type or *UBE3A*^−/−^ H1299 cells expressing UBE3A or the enzymatically dead C843A mutant were subjected to uPA activity analysis. Data are presented as mean ± SD, one-way ANOVA, with Bonferroni post-hoc test, three independent experiments. * *p* < 0.05, significant difference; ** *p* < 0.01, very significant difference; NS, no significant difference.

**Figure 3 ijms-21-04231-f003:**
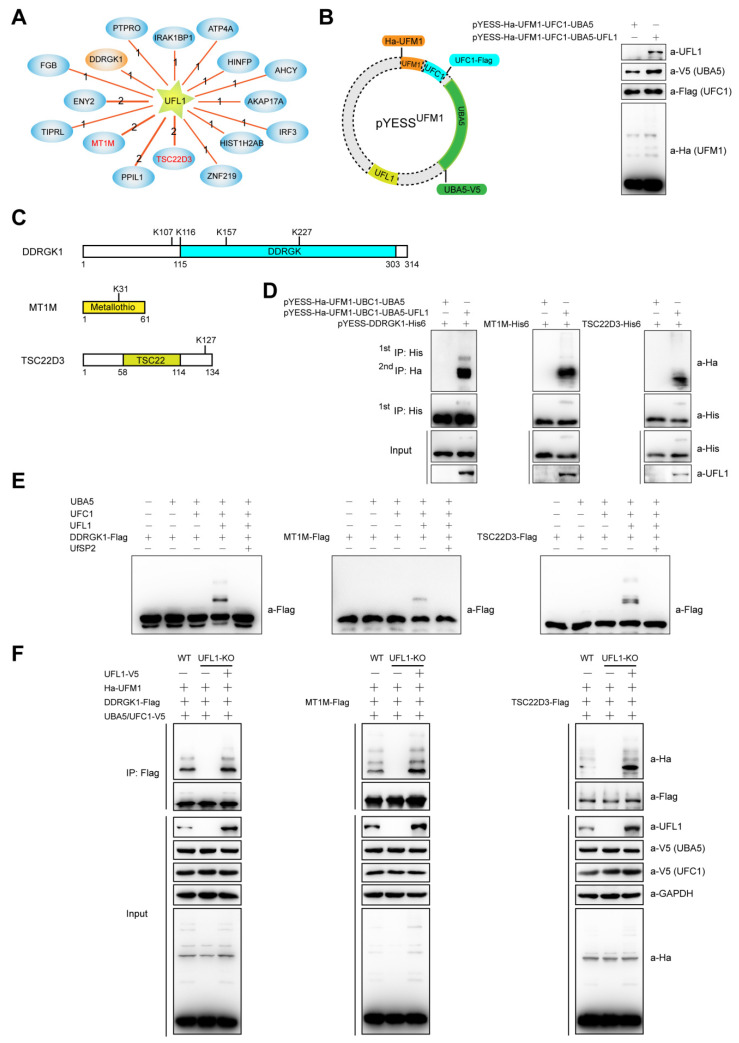
YESS^UFM1^, YESS applied to identify and characterize substrates for UFL1-mediated ufmylation. (**A**) UFL1-interacting proteins screened by Y2H with human UFL1 as the bait. (**B**) Reconstituting the *E. coli* ufmylation system by transforming YESS^UFM1^ vector that expressing UFM1 (Ha-UFM1), E1 (UBA5-V5), E2 (UFC1-Flag) and E3 (UFL1) into BL21 cells and the expression of these proteins was detected by immunoblotting assay. +, transfected with indicated plasmid; −, without transfected with indicated plasmid. (**C**) A schematic illustration of the ufmylation sites on DDRGK1, MT1M and TSC22D3 identified in *E. coli* ufmylation system as revealed by mass spectrum analysis. (**D** and **E**) Substrates validation. UFL1-interacting DDRGK1, MT1M and TSC22D3 were individually subjected to ufmylation assay in *E. coli* (**D**) or in vitro (**E**). (**F**) Detection the ufmylation of DDRGK1, MT1M and TSC22D3 in wild type or *UFL1**^−FL^* HEK293T cells expressing empty vectors or UFL1. −, *UFL1* gene knoctout allele.

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
