# Peer review of "An Integrative Synthetic Biology Approach to Interrogating Cellular Ubiquitin and Ufm Signaling"

_ijms, 2020, doi:10.3390/ijms21124231_

Round 1

Reviewer 1 Report

In their manuscript, Li et al. present an approach to identify and investigate substrates for post-translational modifications. It starts with the well-established yeast-two-hybrid screening to detect putative interaction partners of the respective modifying enzyme. Subsequently, an elaborate and clever genetic approach allows one to investigate, if obtained hits are substrates of the modifying enzyme, in a straightforward manner. Remarkably, this analysis is done in parallel in Escherichia coli and mammalian cells.

As a proof of principle, the authors used their approach (which they refer to as YESS) to identify substrates of the ubiquitin E3 ligase UBE3A and the Ufm1 E3-ligase UFL-1. In the case of UBE3A, they were able to confirm three known substrates as well as to identify one new one. In the case of UFL-1, they confirmed one known substrate and identified two new ones. Their integrative approach also involves mass spectrometry to identify the sites of modification.

This is nice work, and deserves publication. The English language, however, should be improved before publication. In particular, very often plural is used were singular would be appropriate.

E.Coli should be changed to E. coli throughout.

Line 152: Change “… both pYESSUB-UBE3A and pYESS-UBE3A-PIPs were co-transfected into E.Coli …” by “… both pYESSUB-UBE3A and pYESS-UBE3A-PIPs were co-transformed into E. coli …

Author Response

Reviewer 1

Comments and Suggestions for Authors

In their manuscript, Li et al. present an approach to identify and investigate substrates for post-translational modifications. It starts with the well-established yeast-two-hybrid screening to detect putative interaction partners of the respective modifying enzyme. Subsequently, an elaborate and clever genetic approach allows one to investigate, if obtained hits are substrates of the modifying enzyme, in a straightforward manner. Remarkably, this analysis is done in parallel in Escherichia coli and mammalian cells.

As a proof of principle, the authors used their approach (which they refer to as YESS) to identify substrates of the ubiquitin E3 ligase UBE3A and the Ufm1 E3-ligase UFL-1. In the case of UBE3A, they were able to confirm three known substrates as well as to identify one new one. In the case of UFL-1, they confirmed one known substrate and identified two new ones. Their integrative approach also involves mass spectrometry to identify the sites of modification.

This is nice work, and deserves publication. The English language, however, should be improved before publication. In particular, very often plural is used were singular would be appropriate.

R1: We are truly appreciated the reviewer’s encouraging comments and generous support. The language of our manuscript has been revised carefully, and whether to use of singular or plural were carefully checked and adjusted appropriately. ( For example, Introduction part, Paragraph 1, line 18, ‘percent’ change to ‘percentages’ )

E.Coli should be changed to E. coli throughout.

R2: E.Coli was changed to E. coli throughout the manuscript as the reviewer’s suggestion.

Line 152: Change “… both pYESSUB-UBE3A and pYESS-UBE3A-PIPs were co-transfected into E.Coli …” by “… both pYESSUB-UBE3A and pYESS-UBE3A-PIPs were co-transformed into E. coli …

R3: This sentence has been changed as the reviewer’s suggestion.

Finally, on behalf of all authors of this manuscript, we would like to thank the reviewer for all the expert advice and kind suggestions that have helped improve our work tremendously!

Reviewer 2 Report

The authors set up a new approach (YESS) for finding E3 ligase substrates and characterize their PTM. We are aware of such new techniques and the authors proved that their approach may be useful, at least for a first screen, as they have found new substrates for 2 E3 ligases. The main concerns I do have is that that the authors present YESS as a widely expandable approach. Second me this is far from being the case and the authors did not discuss enough about the limitations of YESS. Indeed, YESS (at least this first version) seems limited to a discrete number of E3 ligases (YESS excludes the multimeric RING E3s, the RING E3 with unknown or poorly characterized E2s, the targets (and the E3s) that do not express correctly in bacteria, etc.). They should carefully discuss these limitations and may also suggest potential improvements for future large scale use of YESS.

Major comments:

  1. One main limitation of the YESS approach is that a single E2 can be used at the time and that Ub E3 ligases generally can cooperate with several E2 enzymes, which can modify the type of Ub chains built but also the targeted substrate. Some E3s even need the presence of 2 E2 enzymes for efficient polyUb chain synthesis. This is not a problem with the E3 ligases tested in this article as E6AP/UBE3A belongs to the HECT family that possesses intrinsic catalytic activity, but HECT represent a minority of E3 ligases. I also agree that UBCH7 is the main E2 for UBE3A/E6AP but other E2s exhibit some activity, at least in vitro (see Ronchi et al JBC 2013). For RING E3 ligases, which represent the vast majority of E3s, the choice of the E2 is crucial and is generally unknown or poorly characterized (in vitro only in most cases). Most of the RING E3s are versatile and use different E2s for different purposes. Even the PTM may be different depending on the E2 present. This means that the YESS approach may be limited to a restricted number of E3s (HECT) or to a limited subset of substrates if RING E3s are addressed. Moreover and as stressed by the authors, the multimeric RING E3 ligases are also excluded from the current version of YESS. These limitations need to be discussed in deep and clearly stated.
  2. A second main limitation of this approach is due to the use of coli itself. This has been partially acknowledged by the authors but this aspect of the discussion should be expanded. In bacteria, the production of conformationally-correct proteins is more an exception than the rule. Ubiquitination can thus occur due to the exposure of protein surfaces that are buried in native proteins and may change an E3 partner to an E3 substrate or at least create non-physiological sites of Ubiquitination. This is highlighted by the data presented in Fig. 1C where some sites found in bacteria are not present in HEK293 cells. I agree that co-expression of the target and the E3 in bacteria may partially circumvent this problem for some proteins but this has to be verified for every new target discovered. In addition, the hits that were negative in bacteria (IL24, MCM6, etc) have not been tested in HEK293 cells while they might be UBE3/E6AP substrates in mammalian cells because of the oversimplified bacteria model (conformational problems, lack of phosphorylation, absence of a partner, etc). Thus, I would be extremely cautious with the statement in l. 200-203. The fact that most Ub sites observed with 4 proteins fit between bacteria and mammalian cells does not mean this is a rule for future studies and that proteins produced in bacteria are equivalent to the ones produced in mammalian cells. As an example, some proteins need to be phosphorylated (or dephosphorylated) before being ubiquitinated. This is obviously not possible in bacteria. Again, YESS is an attractive approach but with limitations. In summary, for future studies, validating the targets and confirming the nature of the PTM and its localization in mammalian cells will be mandatory for validating the YESS approach, the latter being a first screen dedicated to easy-to-handle proteins and not a stand-alone technique. These limitations must be clearly stated and discussed.

Minor comments:

  1. In the introduction section (l. 39-39), I agree that E3 ligases dictate the specificity but this not true in most cases for the Ub code, as the majority of E3 enzymes lack enzymatic activity. It is the E2 interacting with the E3 that dictate the type of Ub chains (numerous publications and review can be cited for that aspect)
  2. 21: Other approaches have been described in the literature and should be cited (e.g. Burande, Mol Cell Proteomics 2009).
  3. 91-92: One should read: “… but not necessarily be its substrates, …”
  4. 2I: The arrows in the small picture on top of this panel may be reversed: increased Hoxd4 and decreased Fdf8 when high RA is present, as shown in the graphs.
  5. 277: One should read; “… had a little effect on that of MT1MK31R “
  6. English needs to be corrected, possibly by a native-English speaker

Author Response

Comments and Suggestions for Authors

The authors set up a new approach (YESS) for finding E3 ligase substrates and characterize their PTM. We are aware of such new techniques and the authors proved that their approach may be useful, at least for a first screen, as they have found new substrates for 2 E3 ligases. The main concerns I do have is that that the authors present YESS as a widely expandable approach. Second me this is far from being the case and the authors did not discuss enough about the limitations of YESS. Indeed, YESS (at least this first version) seems limited to a discrete number of E3 ligases (YESS excludes the multimeric RING E3s, the RING E3 with unknown or poorly characterized E2s, the targets (and the E3s) that do not express correctly in bacteria, etc.). They should carefully discuss these limitations and may also suggest potential improvements for future large scale use of YESS.

   R1: First of all, we would like to thank the reviewer for your professional and thoughtful suggestions. The concerns about our approach (YESS) were discussed in the discussion part of our manuscript. (Discussion part, Paragraph 4)

Major comments:

  1. One main limitation of the YESS approach is that a single E2 can be used at the time and that Ub E3 ligases generally can cooperate with several E2 enzymes, which can modify the type of Ub chains built but also the targeted substrate. Some E3s even need the presence of 2 E2 enzymes for efficient polyUb chain synthesis. This is not a problem with the E3 ligases tested in this article as E6AP/UBE3A belongs to the HECT family that possesses intrinsic catalytic activity, but HECT represent a minority of E3 ligases. I also agree that UBCH7 is the main E2 for UBE3A/E6AP but other E2s exhibit some activity, at least in vitro (see Ronchi et al JBC 2013). For RING E3 ligases, which represent the vast majority of E3s, the choice of the E2 is crucial and is generally unknown or poorly characterized (in vitro only in most cases). Most of the RING E3s are versatile and use different E2s for different purposes. Even the PTM may be different depending on the E2 present. This means that the YESS approach may be limited to a restricted number of E3s (HECT) or to a limited subset of substrates if RING E3s are addressed. Moreover and as stressed by the authors, the multimeric RING E3 ligases are also excluded from the current version of YESS. These limitations need to be discussed in deep and clearly stated.

    R2: Thanks for the reviewer’s thoughtful suggestion. The limitations that you’re mentioned were added to the discussion part of our manuscript. (Discussion part, Paragraph 4, line 1-13)

  1. A second main limitation of this approach is due to the use of coli itself. This has been partially acknowledged by the authors but this aspect of the discussion should be expanded. In bacteria, the production of conformationally-correct proteins is more an exception than the rule. Ubiquitination can thus occur due to the exposure of protein surfaces that are buried in native proteins and may change an E3 partner to an E3 substrate or at least create non-physiological sites of Ubiquitination. This is highlighted by the data presented in Fig. 1C where some sites found in bacteria are not present in HEK293 cells. I agree that co-expression of the target and the E3 in bacteria may partially circumvent this problem for some proteins but this has to be verified for every new target discovered. In addition, the hits that were negative in bacteria (IL24, MCM6, etc) have not been tested in HEK293 cells while they might be UBE3/E6AP substrates in mammalian cells because of the oversimplified bacteria model (conformational problems, lack of phosphorylation, absence of a partner, etc). Thus, I would be extremely cautious with the statement in l. 200-203. The fact that most Ub sites observed with 4 proteins fit between bacteria and mammalian cells does not mean this is a rule for future studies and that proteins produced in bacteria are equivalent to the ones produced in mammalian cells. As an example, some proteins need to be phosphorylated (or dephosphorylated) before being ubiquitinated. This is obviously not possible in bacteria. Again, YESS is an attractive approach but with limitations. In summary, for future studies, validating the targets and confirming the nature of the PTM and its localization in mammalian cells will be mandatory for validating the YESS approach, the latter being a first screen dedicated to easy-to-handle proteins and not a stand-alone technique. These limitations must be clearly stated and discussed.

  R3: Thank you for the reviewer’s thoughtful suggestion. The limitations that you’re mentioned were added to the discussion part of our manuscript. We are very appreciated that the reviewer’s suggestions for the improvement of future studies. (Discussion part, Paragraph 4, line 13-24) 

Minor comments:

  1. In the introduction section (l. 39-39), I agree that E3 ligases dictate the specificity but this not true in most cases for the Ub code, as the majority of E3 enzymes lack enzymatic activity. It is the E2 interacting with the E3 that dictate the type of Ub chains (numerous publications and review can be cited for that aspect)

    R4: “The specific interaction between E3 ligases and their respective  substrates is the major factor dictating the specificity and the codes for Ub signaling” has been changed to “The specific interaction between E3 ligases and their respective substrates is the major factor dictating the specificity for ubiquitination modification, while the E2 interacting with the E3 that dictate the type of Ub chains.”. (Introduction part, paragraph1, line 13-16)

The citations have been added in the introduction part (see reference 11-12).

  1. 21: Other approaches have been described in the literature and should be cited (e.g. Burande, Mol Cell Proteomics 2009).

   R5: The literature mentioned has been added in the discussion part,  paragraph1, line 4-7. (reference 47).

  1. 91-92: One should read: “… but not necessarily beits substrates, …”

   R6: The error has been corrected.

  1. 2I: The arrows in the small picture on top of this panel may be reversed: increased Hoxd4 and decreased Fdf8 when high RA is present, as shown in the graphs.

    R7: We corrected this mistake in Fig.2I.

  1. 277: One should read; “… had a little effect on that of MT1MK31R “

    R8: This error has been corrected.

  1. English needs to be corrected, possibly by a native-English speaker

     R9: The language of our manuscript has been revised carefully.